# Replication Stress, Genomic Instability, and Replication Timing: A Complex Relationship

**DOI:** 10.3390/ijms22094764

**Published:** 2021-04-30

**Authors:** Lina-Marie Briu, Chrystelle Maric, Jean-Charles Cadoret

**Affiliations:** Université de Paris, CNRS, Institut Jacques Monod, F-75006 Paris, France; lina-marie.briu@ijm.fr (L.-M.B.); chrystelle.maric@ijm.fr (C.M.)

**Keywords:** replication timing, replication stress, genomic instability, cancers

## Abstract

The replication-timing program constitutes a key element of the organization and coordination of numerous nuclear processes in eukaryotes. This program is established at a crucial moment in the cell cycle and occurs simultaneously with the organization of the genome, thus indicating the vital significance of this process. With recent technological achievements of high-throughput approaches, a very strong link has been confirmed between replication timing, transcriptional activity, the epigenetic and mutational landscape, and the 3D organization of the genome. There is also a clear relationship between replication stress, replication timing, and genomic instability, but the extent to which they are mutually linked to each other is unclear. Recent evidence has shown that replication timing is affected in cancer cells, although the cause and consequence of this effect remain unknown. However, in-depth studies remain to be performed to characterize the molecular mechanisms of replication-timing regulation and clearly identify different *cis*- and *trans*-acting factors. The results of these studies will potentially facilitate the discovery of new therapeutic pathways, particularly for personalized medicine, or new biomarkers. This review focuses on the complex relationship between replication timing, replication stress, and genomic instability.

## 1. Introduction

The process of DNA replication enables the faithful and complete duplication of the cell genome in order to transmit a copy in its entirety to daughter cells after cell division. This molecular process must be finely orchestrated and coordinated with other molecular processes in the nucleus to avoid defects in DNA replication, which can be deleterious to the cell. Although the process of DNA replication takes place during the S-phase of the cell cycle, the regulatory mechanisms, including checkpoints, occur during the G2-phase of the previous cell cycle and throughout the G1-phase [1]. These regulatory mechanisms not only determine the replication starting points throughout the genome, but also define when each part of the genome is replicated during the S-phase. This temporal program defines the replication timing (RT) and seems to play a major role in the organization of eukaryotic genomes. With improvements in high-throughput genomic approaches, several correlations have been demonstrated between replication timing and chromatin stability [2]. It is now well known that the deregulation of replication timing and changes in chromatin can lead to the emergence of cancer cells. However, a key question remains unanswered: What is the first event in the process that ultimately leads to genome instability?

In this review, we focus on the different actors involved in replication timing and its correlations with molecular elements in the genome. We also explore the link between the replication program and the mutational landscape in the genome, together with its evolutionary conservation. We then investigate the potential relationship between replication timing and DNA replication stress. DNA replication stress represents alterations in the velocity of ongoing replication forks, resulting in asymmetry in the fork progression, unreplicated regions, and/or fork collapse. We finally discuss how the deregulation of this temporal program affects chromatin stability, examine the links between them, and explore the causes and consequences of changes in the replication timing in the context of tumorigenesis.

## 2. Characteristics of the Replication-Timing Program

For many years, it has been well known that different parts of the metazoan genome are replicated at different time points during the S-phase. Genome replication does not occur randomly and is defined by a very precise sequence of events called the RT program [3]. Given the magnitude of this task, this program is not initiated the moment before it occurs, but at the beginning of the G1-phase with the timing decision point (TDP [1]). Notably, the program is implemented before the onset of the spatial program of replication with the origin decision point (ODP, which defines the position of replication origins along the genome [1]), perhaps indicating the preponderance of the temporal program compared to its spatial counterpart. Moreover, if the implementation of the RT program is disrupted, then the replication of the genome during the S-phase is completely disturbed, with consequences for the stability of the genome itself [1,4]. This sequential nature of replication is important for the cell because it enables coordination with the different factors involved such as nucleotides [5]. It is also crucial for its coordination with different DNA repair systems and its adaptive response to replication stress via translesional polymerases [6]. The temporal program of replication is therefore a coordinator that ensures the complete and perfect duplication of the genome in the allotted time. Another factor that suggests the importance of RT is its robustness. Indeed, RT variations between cells of the same tissue are infinitesimal within an organism and close to 1–2% for the same cell type between different individuals. However, RT can vary by about 50% during differentiation or between different cell types [7,8]. Thus, RT can be regarded as an epigenetic mark [9]. Based on when they are replicated, genome segments can be classified into two types: (i) regions that follow the same global replication timing, which are called CTRs (constant timing regions), and (ii) transition regions called TTRs (timing transition region), which are located between two CTRs. CTRs fall into two categories: (i) early CTRs are molecularly characterized by pronounced GC enrichment and a high gene density, and are globally associated with open chromatin and high transcriptional activity; and (ii) late CTRs are associated with AT-rich, gene-poor regions that are enriched with repeated elements, and they colocalize with closed chromatin structures at the periphery of the nuclear membrane [7].

To date, few molecular elements that regulate RT have been identified. Gene invalidation approaches (by knock-out (KO) or knock-in (KI)) have been used to study the effects of these elements, and it appears that the measured percentage of genome-wide RT changes never exceeds a rather low limit. For instance, depletions of some RT regulators such as RIF1, SUV4-20H, and DNA polymerase θ (Polθ) have been shown to induce around 16%, 15%, and 5% of genome-wide RT changes, respectively [10,11,12]. This indicates that either compensatory mechanisms exist to maintain this crucial program or a large number of regulatory factors cannot be identified because the induced RT changes are too drastic for the cell to survive. This could explain why laboratories studying RT have never observed changes in values beyond a limited threshold and why the detection of RT molecular regulators remains challenging. Despite these difficulties, several *cis* and *trans* classes of RT regulatory factors have been described. First, a strong origin associated with an active promoter in a given region enables its earlier replication in the S-phase [13]. Then, the ERCEs (early replication control elements) associated with certain open epigenetic marks and some other factors allow for the early replication of the replication domains connected to them [14]. Knowing the strong correlation between RT and 3D genome interactions, we can reasonably question whether this effect may be directly caused solely by the destabilization of these interactions. Epigenetic mechanisms may also be associated with RT regulation, as demonstrated by several findings such as the inactivation of the X chromosome, whereby replication of one allele occurs early and that of the other occurs late in the S-phase in female cells [15]. In addition, the H4K20Me3 mark is associated with the control of late-origin activation [11]. These examples are not exhaustive (reviewed in [16]). Finally, the category of factors acting in *trans* also regulates origin activation at a specific time point during the S-phase. The best known thus far is RIF1, whose role in the control of RT is conserved from yeast to humans [10,17,18,19,20]. Pol θ also plays a role in the domains replicated early in the S-phase [12]. It is possible that the helicase domain of this translesional polymerase destabilizes G-quadruplexes (G4), an essential element for the functionality of replication origins [21]. Finally, in yeast, the dimerization of the transcription factors FKH1 and 2 allows them to cluster several early origins for mutual activation [22].

In the last few years, chromatin conformation capture approaches and RT profile studies have shown that there is a strong link between RT, 3D interactions, and the organization of the genome [16,23]. CTRs consist of one or several replication domains (RDs), which replicate at similar times. They are separated into two groups corresponding to the A compartment, which is the active form and localizes with early replication domains, and the B compartment, which represents the inactive form and is associated with late replication domains. A and B compartments likely represent euchromatin and heterochromatin compartments, respectively [16,23,24,25]. RDs are composed of one or several topologically associated domains (TADs). Some of them are located near compartment boundaries, corresponding to RT switching [23,26]. Thus, TADs and RDs, which seem to be intimately linked, can be considered arbitrary units of the organization and structure of a genome [23]. This link appears to be very strong: for example, during cellular differentiation, several studies have shown that the interactions of TADs co-evolve with RT in parallel with associations with the transcriptional program [16,23,26]. Other evidence clearly demonstrates this intimacy between the 3D genome organization and RT [27]. Moreover, there is an interesting parallel between the TDP and the organization of chromatin domains. Interactions between different domains that disappear during mitosis have been observed to re-establish during the same TDP time window [27]. Although these observations do not clarify whether the 3D organization of the genome controls the establishment of RT or vice versa, the same study showed that inter-TAD interactions were not necessary for the establishment of the RT program. Therefore, these two processes are regulated in parallel by common molecular elements such as RIF1 [20].

The organization and structure of the genome are known to be crucial for the identity and fate of the cell. This is similarly the case for RT, as studies using abortive approaches of different factors that regulate it and treatments with different drugs have never shown more than a limited percentage of RT changes [10,11,12,28,29,30]. One possible explanation is that a degree of change beyond this limit is so high that it is catastrophic for the cell, as it must significantly disturb other molecular processes in the nucleus. This hypothesis is supported by studies that have detected a correlation between the disruption of RT and several diseases such as Fragile X syndrome and certain cancers [31,32]. Therefore, RT is now considered one of the “primary functions” of the nucleus. This program is so important that it seems to have been selected during evolution. Studies in yeast of the genus *Saccharomyces* have shown that the RT program is fairly well conserved between different species [33]. Furthermore, a very thorough study was carried out on 10 species of the genus *Lachancea*, covering the continuous evolution of their genomes [34]. The results suggest that the RT program evolves at the same rate as protein-coding sequences and the genome structure [34]. However, the evolution of these coding sequences does not dictate the evolution of the RT program. Indeed, for this genus, the disappearance of some origins of replication and the appearance of new ones have modified RT in different species. However, why and how these origins appear and disappear during evolution is still unknown. Two studies that compared the RT program in human and mouse cells showed very strong evolutionary conservation [24,25]. As previously indicated, there is a strong correlation between the replication timing of different domains and the GC content. However, these studies show that the evolution of RT is independent of the evolution of the GC content. Moreover, the RT is conserved despite different chromosomal rearrangements between these two mammalian genomes. Thus, there is a mechanism selected during evolution that has resulted in the conservation of RT between different mammalian species. Once again, these results show the importance of RT and its key role in structuring the genome.

Therefore, the RT program appears to be a robust nuclear mechanism. Nonetheless, as we describe below, variations in the type of mutational events appear to correlate with the RT of the associated replication domains.

## 3. Connections between the Replication Timing Program and Mutational Events

The “primary mission” of replication is to completely duplicate the genome with flawless fidelity. However, there are several constraints. First, the time allowed for this duplication is limited (S-phase only), which necessitates a fine-tuned coordination of this process. Second, the replicative machinery can encounter several obstacles at the DNA and chromatin levels that can slow down or stop the process. There are several types of obstacles: (i) secondary DNA structures or the binding of non-histone proteins to chromatin or RNA–DNA hybrids such as R-loops, (ii) the level of chromatin compaction, and (iii) interference with the transcriptional machinery. This list is not exhaustive and is reviewed elsewhere [35]. All of these obstacles generate replication stress and, consequently, some degree of chromosomal instability. The affected cells will nevertheless set up different repair systems to ascertain the transmission of complete and faithfully copied genetic material to the daughter cells. As with any system, this one is not entirely perfect, and variations, called mutational events, will remain in the genomic sequence. However, the nature and number of these mutations are not random in the genome; there is a connection with the temporal program of replication.

Mutational events fall into three main categories: (i) base substitutions, (ii) copy number variations, and (iii) chromosomal rearrangements. The influence of RT differs between these types of mutational events (Figure 1).

In 2009, one of the first studies on this topic showed that the number of SNPs (single-nucleotide polymorphisms) was not homogeneous throughout the human genome [36]. Indeed, the results of this study showed a greater accumulation of SNPs in late-replicating regions, independent of their GC content, the mutational effects of recombination events, and the presence or absence of genes. This indicates that a conflict between replication and transcription is not at the origin of this event. In addition, no base substitutions were observed to be particularly enriched. Furthermore, there were no notable differences among transversion or transition mutations. Nonetheless, there seems to be a bias depending on whether SNPs are on the leading or lagging DNA strand [37]. Other work has confirmed these results in humans [38,39]. The accumulation of base substitutions in late regions is not unique to the human genome; the same results have been found in the yeast *S. cerevisiae* and in several species of flies of the genus *Drosophila* [40,41]. Finally, the same observations have been made in cancer cells [42,43,44,45]. Some works suggest that early-replicating regions containing base substitutions are more likely to be repaired than late regions before the end of the S-phase [36,46]. Other studies suggest that repair systems used late in the S-phase are less accurate than those used in the early stage [47,48]. Finally, in metazoans, heterochromatin located at the periphery of the nucleus at the level of the nuclear envelope may be more exposed to exogenous stress than euchromatin located deeper inside the nucleus [49]. Heterochromatin therefore potentially has more base substitutions than euchromatin, as heterochromatin is generally replicated in the late S-phase [49].

While there is a general consensus about base substitution rates and late replication among different eukaryotes, there are no common rules regarding CNVs and replication timing. A study on iPS cells indicates that amplifications are associated with early-replicating regions, while deletions are associated with later ones [50,51]. Another study investigated the nature and the distribution of these CNVs as a function of RT in 26 cancer lines [52]. Once again, amplifications and deletions were associated with early- and late-replicating regions, respectively. However, the rule is the opposite in *Drosophila melanogaster*: in other words, losses are associated with early-replicating regions, while amplifications are associated with late-replicating regions [53].

What could explain these differences between eukaryotes? Unfortunately, it is difficult to answer this question because studies on other model organisms are lacking. CNVs seem to appear following the stalling of the fork and its destabilization [54]. Data from our team showed that CNVs are enriched within pausing sites of replicative DNA polymerases (unpublished data). To counteract fork collapse, cells use recombination systems to restart replication. These recombination systems favor domain amplifications and early-replicating regions because these regions have more replication modules that can potentially be partners for these recombination systems. These remain working hypotheses, and nothing has yet been proposed as an explanation for the deletions and the mechanisms involved in *Drosophila*. Thus, further research on this topic is required.

Studies on the positions of chromosomal translocations in mouse and human genomes have shown that these translocations are enriched in regions that undergo early replication [24]. Studies by Chiarle et al. and Zhang et al. showed that translocations were precisely localized at the transcriptional start sites (TSSs) of active genes [55,56]. As described previously, most active genes are replicated early, hence the correlation between translocations and RT. Translocations are a consequence of DNA double-strand breaks (DSBs), which may be due to replication stress. This stress can be caused by (1) collisions between transcriptional and replicative machineries [57], (2) the presence of topological stress that is generated by transcription and poorly managed by the replication process [58], or (3) the presence of R-loops. All of these mechanisms are now known to generate replication stress [59].

## 4. Specific Genomic Regions More Prone to Replication Stress during the S-Phase

Each step of the temporal DNA replication program (early, mid, and late S-phase) seems to be associated with replication stress at specific genomic loci in challenging replication conditions. For different reasons, these specific genomic regions are more prone to replication stress than others and strongly contribute to genomic instability.

The most well-described example is the sensitivity of common fragile sites (CFSs). CFSs are characterized by specific genomic features that are similar to late-replicating regions. They colocalize with very large genes (~80% of CFSs nest in genes with a size greater than 300 kb, whereas medium-sized human genes are ~20 kb), contain long AT-rich sequences (from ~100 bp to several kb), have a paucity of replication origins and are replicated in the late S-phase (e.g., FRA3B, FRA16D; reviewed in [60,61,62]). While these features disturb replication progress at CFSs under normal conditions, molecular defense mechanisms keep them stable. Therefore, cells experiencing DNA replication stress may exhibit mitotic DNA synthesis (MiDAS) [63]. To understand the physiological function of MiDAS and its relationship with CFSs, the genomic sites of MiDAS were mapped in cells treated with aphidicolin (APH, an inhibitor of replicative DNA polymerases). MiDAS sites were observed as well-defined peaks that were largely conserved between cell lines and encompassed all known CFSs. The MiDAS peaks were mapped to large, transcribed, origin-poor genomic regions. In cells that had been treated with APH, these regions remained unreplicated, even in the late S-phase. MiDAS is thus a rescue process that may facilitate the completion of replication in these regions [64]. Nevertheless, replication timing is an important parameter that causes CFS instability. Under replication stress, the replication of CFSs is further delayed, increasing the risk of the incomplete replication of these regions in the S- and G2-phases. This leads to chromosomal instability when cells enter mitosis (i.e., breaks on metaphase chromosomes and the occurrence of chromosomal rearrangements [61]). The phenomenon is accentuated upon APH treatment, which further delays the replication of CFS regions (e.g., more than 10% of FRA3B remains unreplicated after APH treatment [65]). However, it is notable that late replication alone is not sufficient to contribute to CFS fragility, as more than 1% of the human genomic DNA replicates very late, even in G2, without becoming fragile [62,66]. It is currently accepted that CFS fragility is due to the intrinsic characteristics of these regions, but their relative contribution is still debated. This fragility may be explained by the combination of more than one mechanism related to the intrinsic features of CFSs in the context of late replication and replication stress. Thus, three mechanisms have been proposed to explain this specific delay in replication completion at CFSs (reviewed in [60,61,62]). First, late-replicating CFSs map to very long genes whose transcription extends to more than one cell cycle, and several works have shown that this causes transcription–replication machinery collisions that induce the formation and accumulation of R-loops (DNA–RNA hybrids), leading to replication fork stalling and collapse. Second, it has been shown that the AT-rich sequences present at CFSs (CFS-ATs) are prone to forming stable secondary DNA structures such as hairpins, which could constitute barriers that are able to stall the DNA replication machinery [67,68,69]. However, the tissue specificity of CFSs questions the exclusive role of *cis*-acting AT sequences. Furthermore, the impact of these first two mechanisms has been minimized by Letessier et al., who found no differences in the frequency of slowed or stalled forks between FRA3B (the most active CFS in human lymphocytes) and the bulk genome, with or without APH treatment [70]. This work established that the fragility of FRA3B does not depend on fork slowing or stalling but on a paucity of initiation events. This is the third mechanism that can explain the specific delayed replication of CFSs. Since CFSs are flanked by origins that fire in the mid-S-phase and are very long regions devoid of replication origins (e.g., FRA3B covers a 700 kb region with almost no origins), they are passively replicated by long-traveling forks. These forks will be more affected by slowing down than those traveling shorter distances. Moreover, the absence of origins and, importantly, of dormant origins in these regions does not allow for the rescue of replication, especially upon replication stress. Recent works that show the transcription-dependent regulation of replication dynamics have complexified the relative role of the different mechanisms involved in CFS fragility [71]. Thus, the risk of CFS breakage is highly increased by replication stress (resulting in replication fork slowing or stalling) that occurs in these regions, combined with the above-described multifactorial genomic context and late replication.

Another subset of fragile sites has been identified: early-replicating fragile sites (ERFSs). Barlow et al. found that high hydroxyurea (HU) doses (10 mM) induced the slowing and collapse of the replication fork during early replication at specific sensitive genomic sites [72]. In contrast to CFSs, ERFSs are nested in intergenic regions of highly transcribed gene clusters, composed of GC-rich sequences and enriched in repetitive elements. These sites are located near replication origins in regions with a high density of replication initiation events [60,61,62,72]. The major difference between these two types of fragile sites is that ERFSs replicate in the early S-phase and, upon replication stress, are associated with DNA breaks in the S- and G2-phases of the cell cycle. Two mechanisms have been proposed to explain the fragility of ERFSs. On one hand, these regions would be prone to forming secondary DNA structures due to their richness in GC, which can stall the DNA replication machinery [62]. On the other hand, it has been observed that the high transcriptional activity in these regions is associated with an increase in replication origin firing, resulting in replication–transcription conflicts and the formation of DNA–RNA hybrids. These events could thus lead to a decrease in the replication fork velocity, followed by replication fork stalling and collapse [73]. In cancer cells, abnormally high transcriptional activity could be the result of oncogene overexpression.

Taken together, these results implicate replication timing as a key parameter of fragile site instability, particularly via the complex relationship between the timing of origin firing and the transcription dynamics of these regions. These interconnections are very important for the coordination of both replication and transcription processes and are thus critical for replication fidelity. To summarize, the impact of replication timing on CFS fragility seems to arise from a failure to complete replication, largely due to a paucity of origins in an end-phase replication context. ERFS fragility appears to arise from an increase in fork collapses due to the discoordination between high transcriptional activity and numerous replication initiation events that are specific to the beginning of the S-phase.

Furthermore, some studies suggest that timing transition regions also make genomic regions more prone to replication stress sensitivity. TTRs are located between early- and late-replicating domains that primarily replicate from the early-mid to late S-phase. These very large areas, which contain no or very few replication origins, are characterized by a progressive change in the replication timing and the GC content [74]. TTRs are passively replicated by a single unidirectional fork that initiates from an early-replicating origin and progresses until reaching the replication fork coming from an adjacent later origin [74,75]. This specific replication pattern could explain their sensitivity to replication stress. Then, a consequence of these passively replicated regions could be an increase in genomic instability due to the unresolved stalling of the replication fork. The sequence content of TTRs could also be a source of replication stress [76]. One study showed that, in addition to a change in the GC content, these regions contain many repetitive sequences (e.g., clusters of Alu elements; polypurine/polypyrimidine tracts; di-, tri-, and tetranucleotide short tandem repeats), which can potentially adopt non-B DNA structures [77]. Many studies have shown that non-B DNA structures are able to arrest replication forks in vitro and in vivo (reviewed in [78,79]). We can thus hypothesize that these types of structures may affect the progression of the replication machinery in TTRs and induce replication stress, although to the best of our knowledge, this phenomenon has not been explored in human cells.

Thus, specific regions of the genome are more sensitive to replication stress than others due to particular genomic or genetic contexts. Importantly, the time of the replication (early, mid, or late) plays a role in this sensitivity. Transcription–replication conflicts and secondary DNA structures seem to be the main sources of replication stress in these fragile genomic regions. Other sources of replication stress can occur during the S-phase and disturb replication dynamics. Potential links between these factors and replication timing are discussed in the next section.

## 5. Other Sources of Replication Stress during Replication and Their Potential Relation-Ship with Replication Timing

### 5.1. Variations in DNA Components

In a physiological context, the cellular dNTP pool is finely regulated with specific concentrations for each nucleotide [80]. Reduction or variation in these dNTP (deoxyribonucleotides triphosphate) levels is one of the sources of replication stress (Figure 2) [81].

Indeed, an appropriate supply of dNTPs during the S-phase is critical for DNA synthesis dynamics. During replication, the dNTP pool is maintained by the ribonucleotide reductase (RNR) enzyme, which catalyzes the reduction of ribonucleoside diphosphates in deoxyribonucleoside diphosphates [82,83]. Studies in yeast have shown that artificial changes in the dNTP level can affect both the replication fork velocity and the initiation rate [84]. In this study, the fork speed in cells treated with HU (an RNR inhibitor) underwent about a 10-fold reduction as soon as the dNTP pool dropped below a critical level. In contrast, the deletion of *SML1* (a gene encoding an allosteric repressor of RNR) induced a 66% increase in fork speed. Interestingly, this deletion caused a 2.5-fold increase in dNTP levels, indicating that replication fork velocity is sensitive to changes in dNTP levels. Studies in human cells have shown similar results [85,86,87], particularly in the context of early cancer stages [88]. In a model of the aberrant activation of the Rb-E2F pathway by cellular oncogenes, the overexpression of cyclin E in human fibroblasts resulted in a dramatic reduction in the dNTP pool, which was associated with a decrease in fork speed. This replication stress was rescued by supplying exogenous nucleosides. Thus, this low-nucleotide pool in oncogene-expressing cells resulted from a failure to activate nucleotide biosynthesis pathways.

Interestingly, both studies reported that a reduction in dNTP levels also affected the density of replication initiation events. However, the origin response appears to be different among species. In yeast, the decrease in the dNTP pool induced by HU treatment resulted in a 25-fold reduction in origin firing [84]. Conversely, the decrease in the dNTP pool linked to cyclin E overexpression in humans is associated with a shortened interorigin distance (i.e., an increase in origin activation) that can be rescued by supplying exogenous nucleosides [88]. This suggests that with chronic exposure to a reduced dNTP pool in cells overexpressing cyclin E for 2–4 weeks, an increase in the density of origins is reduced during the G1-phase to compensate for DNA replication stress. One important nuance is that the pattern of initiation events seems directly related to the replication fork speed rather than to the nucleotide pool [89,90]. Thus, dNTP level changes may indirectly affect the number of active origins via their impact on fork speed. However, the precise hierarchical organization between dNTP levels, fork speed, and origin usage remains unclear. Thus, the dNTP pool is a key modulator of replication dynamics and may play a role in the regulation of DNA RT, although this has never been explored (Figure 2). Since an altered nucleotide pool can modulate origin activation and fork speed, this could result in global changes in the RT program, with advanced or delayed replication of some genomic regions. This is worth analyzing since alterations in the replication-timing program are correlated with tumorigenesis. The relative proportion of each dNTP (dATP, dCTP, dGTP, dTCP) in normal and cancer human cells has been previously established [80]. In normal cells, the four dNTPs are present at different concentrations in vivo, with the dGTP concentration being much lower than the others. Conversely, in cancer cells, the whole dNTP pool is substantially increased, although the dGTP level remains low compared to the levels of the other dNTPs. A more recent study measured the relative levels of the four dNTPs in mouse cells during the S-phase and found that the AT/GC deoxynucleotide ratio significantly increased during the progression of the S-phase [91]. The mechanism responsible for higher levels of dATPs and dTTPs in the late S-phase is not known yet. To our knowledge, no direct relationship has been shown between GC-rich early-replicating regions, AT-rich late-replicating regions, and the dNTP ratio. This remains to be explored in detail. Thus, the composition of the dNTP pool is not uniform and varies from the early to late S-phase. To further the current understanding of this process, it would be informative to measure the relative levels of the four dNTPs and the RNR activity during the S-phase in human normal and cancer cells in order to answer several questions. Are dNTP levels and their relative proportions the same toward the beginning, the middle, and the end of the S-phase? Does the same RNR enzyme pool produce dNTPs throughout the whole replication process? Does this then affect the enzyme efficiency at the end of replication, thus potentially contributing to a reduced or an imbalanced dNTP pool? Answering these questions will allow for the evaluation of potential S-phase time points at which the genome is more sensitive to replication stress in challenging conditions and that could contribute per se to genomic instability and promote the development of early stages of cancers.

Ribonucleotides or ribonucleotide-induced DNA breaks can impede replication fork progression and lead to replication stress (Figure 2). Ribonucleotide insertions during the replication process are a general feature of eukaryotic genomes. Human polymerase δ (Polδ) has been shown to stably incorporate one ribonucleotide per approximately 2000 nucleotides. This rate predicts that more than a million ribonucleotides are incorporated into the genome per replication cycle [92]. Furthermore, at physiological concentrations of nucleotides, human polymerase ɛ (Polɛ) also readily inserts ribonucleotides. As almost half of inserted ribonucleotides escape proofreading, ribonucleotide incorporation by Polɛ is also likely a significant event in human cells [93]. It has also been shown that the leading-strand DNA Polɛ incorporates more ribonucleotides than the lagging-strand Polδ in the yeast genome and that the proofreading activity performed by DNA polymerases on ribonucleotides is absent or weak [94,95]. A genome-wide analysis showed that ribonucleotides are not inserted at random in the yeast genome. These events are driven by the nucleotide immediately upstream of the site of incorporation. A deoxyadenosine is found upstream of the most abundant genomic ribonucleic cytosines and guanines, and there is a strong propensity for ribonucleotides to be incorporated into short-nucleotide repeats that initially contain deoxycytosines and deoxyguanines [95]. It is tempting to speculate that the inserted ribonucleotides contribute to the instability of these regions, leading to the expansion or contraction of these short repeats [95]. However, to date, no studies have shown whether ribonucleotides are incorporated at specific time points during the S-phase in human or even yeast genomes. As inserted ribonucleotides are enriched in GC motifs, it is reasonable to hypothesize that these events may be more frequent in GC-rich genomic regions that are more likely to be located in early-replicating domains.

The presence of ribonucleotides may also have structural implications for eukaryotic genomes. RNA is 100,000 times more susceptible than DNA to spontaneous hydrolysis under physiological conditions, which may end in strand breaks. Furthermore, the chromatin assembly taking place during replication as well as the nucleosome positioning may be affected by the presence of ribonucleotides. The nucleosome assembly on the lagging strand could occur before Okazaki fragment maturation, leading to the incorporation of the RNA primer into nucleosomal DNA [96], although it has been shown in vitro that nucleosome–DNA binding is reduced when ribonucleotides are embedded in DNA [97]. Notably, replicative polymerases are not proficient at bypassing ribonucleotides embedded in DNA, which can impede replication fork progression. Thus, paused or stalled DNA polymerases and DNA strand breaks due to the presence of embedded ribonucleotides in the genome may lead to replication stress and to potentially altered replication timing.

### 5.2. Presence of Proteins on DNA or Chromatin

Particular proteins on DNA or chromatin form DNA–protein complexes, which have a direct influence on RT in human cells (Figure 2). For example, the RIF1 protein is bound mainly to late-replicating regions in the normal cell cycle, whereas these same regions undergo earlier RT in RIF1-depleted cells [10,17]. RIF1-deficient cells are very sensitive to replication stress, suggesting a direct role for this protein in the replication stress response. It was shown recently that RIF1 contributes to the recognition of under-replicated or unrepaired sites and is present during replication stress and recovery to promote the cell survival [98]. In contrast to RIF1, Polθ is implicated in the early steps of DNA replication, and its depletion causes delayed replication [12]. It was also recently revealed that ORC density is correlated with replication timing. ORC was observed to preferentially bind to early-replicating chromatin regions and transcription start sites of actively transcribed genes [99]. Furthermore, the proteins DONSON and FANCM are associated with euchromatin replicated early in the S-phase and heterochromatin replicated in the late S-phase, respectively [100]. All of these studies suggest that DNA–protein interactions have a direct role in the establishment of RT that may be modified and/or reinforced in response to replication stress; however, this has not been fully demonstrated to date.

Different histone proteins and chromatin marks may have a more indirect association with the set-up of replication timing. For example, histones H3.3 and H3.1 have been shown to preferentially bind early- and late-replicating chromatin, respectively. H3.3 forms domains in the nucleus that decrease in density during the replication process [101]. Some histone modifications such as methylation or acetylation have also been shown to be associated with RT (Figure 2). 

Therefore, the chromatin regions decorated with transcription-permissive marks are replicated during the early stage of the S-phase, and H3K27me3/H3K9me3-decorated chromatin domains are replicated in the late stage of the S-phase [102]. Furthermore, the H4K20me3 histone mark is associated with late origins and is required to ensure the proper replication of late-replicating regions of heterochromatin [11]. Acetylation of histones H3 and H4 also causes some later-replicating DNA sequences to replicate earlier [103]. Nevertheless, hyper-acetylation of H3K56 in yeast has been shown to induce replicative stress. In this case, the replisome must process a histone configuration that is usually found behind the replication fork, rather than in front of it, which may induce the stalling or even the collapse of the fork [104]. In addition, pericentromeric heterochromatin in mouse ES cells switches from early-mid to late S-phase replication upon differentiation due to an increase in chromatin condensation and a decrease in its acetylation [105]. Furthermore, an increase in the phosphorylation of histone H1 promotes earlier replication [106]. The action of these proteins or that of these chromatin marks may then directly or indirectly influence the RT program and even induce replication stress, but, to our knowledge, this has not yet been shown in human cells.

Some proteins have also been shown to play a role in preventing replication–transcription conflicts. For instance, the depletion of FACT in humans and yeast induced transcription-associated genetic instability and fork progression impairment [107]. Furthermore, the depletion of histone H1 induces replication stress and DNA damage linked to replication–transcription conflicts [108]. The replication-timing program is globally maintained in mouse cells with a 50% reduction in histone H1 levels, with a few regions having a shift in timing; these regions have a signature of early-replicating domains and a loss or a gain of H3K4me1 or H3K4me3 chromatin marks, respectively. In addition, the artificially induced transcription of a long late-replicating gene leads to its fragility. However, transcription can prevent fragility by advancing replication timing to earlier in the S-phase, providing more time to complete replication at these regions [71].

Oncogene activation can also lead to replication stress via various mechanisms. The impact of oncogene expression seems to vary depending on the time point in the RT program, and the associated genomic regions are characterized either by origin enrichment and high transcriptional activity (early-replicating regions) or by a scarcity of origins (TTRs and late-replicating regions). *CCNE1* (cyclin E) and *MYC* oncogenes have been shown to induce the firing of new replication origins within highly transcribed genes in early-replicating domains (Figure 2). These ectopic origins are normally inactivated by the transcription process during the G1-phase of the cell cycle. However, in cancer cells, no intragenic origins are inactivated due to the oncogene-induced shortening of the length of the G1-phase [63]. The firing of these origins within transcribed genes may result in replication–transcription conflicts, the formation of secondary structures such as R-loops, and even DSBs and genomic instability. Alternatively, several studies have linked oncogene activation to decreased origin licensing, which may lead to under-replicated DNA or replication stress due to an increase in the speed of replication forks and the generation of genomic instability [90,109,110]. A reduced number of origins, particularly in TTRs or late-replicating regions, might increase the probability of the stalling of replication forks covering longer distances. Furthermore, replication forks that stall in these regions may persist because of a reduced number of dormant origins (normally activated as backups when forks stall), which may then increase the risk of fork collapse and the generation of DSBs.

### 5.3. Existence of Particular DNA Structures

Given their nature, interspersed or tandem repetitive DNA elements present a challenge for the replication machinery and may result in replication stress. Repetitive DNA elements account for more than half of the human genome and are replicated at specific time points of the S-phase (Figure 2). Euchromatic Alu elements are replicated early; LINE-1 elements, which are associated with AT-rich genomic regions, are replicated throughout the S-phase, with the majority being replicated according to their particular histone marks. Satellite III elements, which constitute the pericentromeric heterochromatin, are replicated exclusively during the mid-to-late S-phase [102]. Furthermore, long stretches of repeated nucleotides such as homopolymeric stretches of dA:dT are susceptible to spontaneous and replication-stress-induced DSBs. For example, these sequences are a causal factor in stalling and breakage at both CFSs and ERFSs in response to HU. Long poly (dA:dT) tracts are not only replication pause sites but also a major contributor to R-loop formation within gene bodies [111]. Due to the presence of these types of repeated nucleotide stretches, certain repetitive elements may adopt non-B DNA conformations that may induce fork stalling [112]. In silico analyses have shown that Alu elements include sequences that are able to form Z-DNA, triplexes, and G4 ([113]; reviewed in [114]). Other retrotransposable elements such as L1PAs and SINE-VNTR-Alu (SVA) also harbor G4 DNA-forming sequences (reviewed in [115]). Besides the implication of their intrinsic sequence in replication stress induction, the mechanism of LINE-1 retrotransposition has been shown to cause replication fork stalling via conflicts between LINE-1 insertion intermediates and the replication fork machinery [116].

Other secondary DNA structures can be formed. Among these, R-loops are abundant RNA–DNA hybrid structures covering up to 5% of mammalian genomes [117]. They form preferentially at regions of head-on collision between transcription and replication (Figure 2). R-loops are enriched at the transcription start and termination sites (TSS and TTS) of highly expressed genes, where replication forks mostly progress in a head-on orientation relative to the direction of transcription. Molecular mechanisms that ensure fork pauses and reinitiation at TTSs may require the tight control of DNA torsional stress, as it is perturbed in topoisomerase1 (TOP1)-deficient cells [118]. The block created by the R-loop in front of the fork increases the supercoiling of the leading strand behind the fork, and TOP1 is required to release the torsion. Thus, TOP1 produces multiple nicks that may lead to genome instability when coupled to replicative stress [96]. In TOP1-depleted cells, DSBs also accumulate at TTSs, leading to persistent checkpoint activation, deposition of γ-H2A.Z on chromatin, and global replication fork slowdown. Fork pausing at the TTSs of highly expressed genes containing R-loops prevents head-on conflicts between replication and transcription and maintains genome integrity in a TOP1-dependent manner [118,119]. A study in yeast revealed that the RRM3 DNA helicase and topoisomerase 2 (TOP2) coordinate replication fork progression and fusion at termination sites throughout the genome [120]. This is also observed in the absence of TOP2 [121]. Further studies in *Xenopus* have demonstrated that TOP2 resolves topological stress to prevent converging forks from stalling during replication termination [122]. R-loop levels are known to depend on the orientation of replication–transcription conflicts. Thus, head-on collisions promote the formation of R-loops, while codirectional collisions resolve them [123]. In yeast, TOP2 confines RNA polymerase II and TOP1 at coding units during the S-phase by counteracting the formation of R-loops at gene boundaries [124]. Interestingly, we have observed that more R-loops are located in early-replicating regions than late-replicating ones (unpublished data), which is consistent with the R-loop enrichment in highly expressed genes. We can then hypothesize that perturbations of mechanisms involved in the pausing/restarting equilibrium of the replication machinery in a head-on context could result in genomic instability and disturb the proper replication of early regions, thereby globally affecting the replication-timing program.

## 6. The Potential Impact of Replication Stress on the Temporal Replication Program

Replication stress and replication timing variations are major changes that can alter the replication process. Each of them has been implicated in the generation of genomic instability and cancer. As described above, replication stress and replication timing seem to be closely related, as the type of replication stress appears to vary depending on the time point of the RT program. Surprisingly, almost none of the available literature has explored the direct relationship between replication stress and changes in the temporal organization of replication, highlighting the complexity of their connections. The activation of new origins and the stalling of replication forks are known consequences of replication stress (reviewed in [81]). Thus, it is tempting to propose that alterations in the replication fork speed (slowing or stalling) and/or replication origin dynamics (increase or decrease in the initiation rate) in many genomic regions delay or accelerate the moment of their replication. This may thus affect the duration of the S-phase and the global RT program of the genome. However, there is no general agreement to this hypothesis yet. In human cells entering replicative senescence, the RT program is globally preserved (less than 0.5% of the genome is slightly affected) during senescence-associated replication stress (slowing of fork rates and activation of dormant origins [125]), suggesting that RT is largely resistant to replication stress. In contrast to this study, recent works have shown that low replication stress induced by aphidicolin had significant effects on the RT of several cell lines (with 1.5–6.5% of the genome significantly affected), resulting in advances and delays in the replication of specific genomic domains [29,30]. Notably, these global values of RT variations in the genome are not very high, suggesting that the impact of replication stress on RT is not drastic. Some studies in yeast showed that the induction of replication stress via the depletion of the nucleotide pool or mutations in cell cycle proteins extended the duration of the S-phase by delaying origin firing. However, this did not affect the order and pace of origin activation (i.e., the delay was proportional to the S-phase duration, maintaining a fixed spacing of origin firing despite slower replication) [126,127]. Finally, these studies suggest that RT is not completely insensitive to replication stress but is highly robust against being extensively altered, presumably due to efficient RT regulators and/or effective compensatory mechanisms activated in response to replication stress (reviewed in [128]).

Alternatively, it is possible that RT changes are induced by the consequences of replication stress and persist throughout several cell cycles because they are not properly resolved, as occurs in cancer cells, for example. It is known that replication stress can lead to genomic instability through events such as DNA breaks (reviewed in [81]) or chromatin modifications [129,130,131], which can then alter the duration and pattern of the replication process during subsequent cell cycles. Furthermore, in addition to replication stress, many other aberrant nuclear mechanisms are altered in cancer cells and may threaten the integrity of the DNA replication process. The links between genome/chromatin instability, aberrations in replication timing, and the contribution of this disequilibrium to cancer progression are discussed below.

## 7. Variations in DNA Replication Timing at Specific Loci or in Genome-Wide Regions Are Observed in Cancers

Several studies have reported abnormalities of the RT program in cancer cells. These RT changes can influence cancer-related genes by affecting one allele, causing asynchronous replication [132,133], or both alleles, resulting in a uniform shift in the replication timing of the gene [32]. For example, comparisons of the replication timing between MCF10A normal human breast cells and the corresponding malignant MCF10CA1 cells have revealed specific RT changes in several cancer-related genes such as *TP53*, *RAD51*, *ATM*, *PTEN*, and *cMYC* in breast cancer cells [32]. Except for *TP53*, all of the other genes shifted toward earlier replication. In addition to loci-specific alterations, RT changes at the genome scale have also been identified (Figure 3) [2,134].

Indeed, whole-genome analyses of 17 pediatric leukemia tumors revealed that 9–18% of the genome of leukemic cells had undergone a change in RT compared to control cells [134]. These variations were distributed throughout the genome on all chromosomes. Most of these leukemic cell RT profiles significantly showed that the majority of the genomic regions replicated earlier, rather than later, in leukemic cells compared to the controls. A more recent study presented comparisons of Repli-Seq data between normal and prostate cancer cells and revealed that RT was largely conserved between the two cell types: only 5.7% of the genome differed in RT, with the number of later-replicating domains exceeding that of earlier ones [2]. These later- or earlier-replicating domains were also distributed across all chromosomes. These whole-genome studies imply that much of the RT program remains unchanged in cancers, highlighting the robustness of this process. This also suggests that perturbations of only a small part of this temporal program may be enough to cause sufficient genome instability to induce or participate in tumorigenesis. This is discussed in further detail later in the review.

## 8. Causes of Replication Timing Changes in Cancers

Despite many efforts to answer this question, the precise cause of the RT changes observed in cancers is unknown. Eukaryotic DNA replication is a highly regulated cellular process orchestrated by both spatial and temporal programs. RT is important for replicating the entire genome before mitosis and optimizing the coordination with the transcriptional process [1]. Many teams have made an effort to identify molecular regulators of this RT program, and this is still a topic of intensive research. Clearly, we can surmise that perturbations of the molecular regulators of the RT program during cancer progression can result in timing variations. However, whether these alterations in RT occur before or at the very early stages of cancer (thus participating in tumorigenesis) or appear later in the timeline (thus being a consequence of genomic instability associated with tumorigenesis) remains unclear.

The deregulation of genes directly involved in the DNA replication process, in either the initiation or elongation steps, can cause alterations in RT. Indeed, we expect that the deregulation of proteins involved in origin licensing during the G1-phase can affect the timing of origin firing during the S-phase. In *Drosophila*, depletion of the ORC2 protein leads to RT aberrations [135]. In contrast, in human cells depleted of ORC1, ORC2, or ORC5, the DNA replication initiation, the duration of the process, and the number of used origins are similar to those in WT cells [136,137]. In a study in budding yeast, the number of mini-chromosome maintenance proteins (MCMs) loaded at an origin directly influenced its firing time [137]. In addition, this study revealed that more MCMs were loaded at early rather than late origins, thus increasing the probability that these origins fire early in the S-phase. Very recently, the same laboratory demonstrated that a reduction in the global cellular pool of MCMs causes significant delays in RT [138]. This strongly suggests that the expression level of MCMs is an important factor in the regulation of the RT program. Similar works need to extend these studies to mammalian cells. Nonetheless, a recent genome-wide analysis of the ORC/MCM distribution performed in human cells has weakened the role of ORC and MCM proteins in the regulation of the RT program [99]. It would thus be informative to observe the RT profiles of human cells with mutations/deregulation in/of *ORC* and *MCM* genes to directly assess their roles in the RT program, especially since *ORC* and *MCM* mutations have been observed in human diseases including cancers (reviewed in [139,140]). Interestingly, the Polθ function, which plays an important role in determining the RT of early-replicating domains [12], seems to be particularly important within the cancer context, as high *POLQ* expression is strongly associated with poor clinical outcomes in several cancers [141,142,143,144,145].

The replication timing and the chromatin state are closely related [16,146]. Indeed, it appears that certain histone marks, DNA methylation marks, and chromatin-modifying enzymes have a role in the regulation of RT in many species, from yeast to mammalian cells. Consistently, chromatin modifications [11,103,147,148,149] and the deregulation of chromatin-modifying enzymes [150,151,152] have been shown to change the RT of genomic regions in different organisms. Interestingly, deregulation of the histone variant H2A.Z or the H4K20me3 mark, which ensure the proper replication of early- and late-replicating origins, respectively [11,149], is observed in several major human cancers [153,154,155,156,157,158], suggesting that RT modifications in these regions due to such epigenetic alterations could contribute to cancer transformation. 

Misregulation of key regulators of RT can also cause changes in this program. Notably, many studies have identified the RIF1 protein as an important player in the maintenance of accurate RT. Its role has also been established in yeast [18,19], mice [10,20], and humans [17]. The depletion of RIF1 results in the global alteration of the RT program, with both early-to-late and late-to-early shifts in very large domains [10,17]. This RT deregulation seems to be due to the remodeling of the nuclear architecture [17,20]. Importantly, the *RIF1* gene is mutated in some breast cancer cell lines [159,160], overexpressed in lung cancers, and positively correlates with poor prognosis in lung cancer patients [161]. These studies raise the possibility that important RT changes through *RIF1* deregulation are involved in tumorigenesis. For many years, the RIF1 protein has been identified as the only molecular regulator with such an impact on the temporal DNA replication program. Recently, one study identified the tumor suppressor *PREP1* as a new regulator of RT in human cells [28]. They found that *PREP1* downregulation in HeLa cells induces a late-to-early shift of 25% of the human genome and is associated with a substantial decrease in the Lamin-B1 level. Some studies have also observed that the lack of PREP1 occurs in a large fraction of human cancers and is involved in tumor development [162,163]. Taken together, these results suggest that RT changes induced by *PREP1* downregulation, through the alteration of Lamin-B1 levels, could be involved in cancer transformation. 

Moreover, the relationship between RT and transcription is very complex. In yeast, no correlation between these two processes has been observed [164]. In contrast, in mammals, many studies have identified a positive correlation between early-replicating regions and transcriptional activity at the gene and genome scales [7,16,165], although the relationship is not systematic since several genes in late-replicating regions are also transcribed [166]. However, a causal link between RT and transcriptional activity remains difficult to characterize. Some work supports the notion that transcriptional activity can impact RT [71,167], but it is not clear whether RT variations are dependent on transcriptional activity per se, or on chromatin changes associated with transcription. In contrast, some studies have found that RT and transcription changes can be independent [14,168]. A recent study explored a novel point of view that could clarify these complex observations. For this purpose, the authors combined ChIP-Seq results for transcriptional factors (TFs) and analyses of transcriptional and RT changes across many human ESC lines that were differentiating toward different lineages [5]. Through these analyses, this study identified TFs whose expression was associated with some early-replicating domains. These results suggest that the correlation between transcriptional and RT changes could result from the regulation of early-replicating regions by combinations of particular TFs rather than the intrinsic transcriptional activity of these regions. Consistently, a study in budding yeast showed that the FKH1 and FKH2 TFs are necessary for the early replication of a large number of origins [22], independent of their transcriptional activities. This suggests that RT may be influenced by TFs that have distinct roles in both transcription and RT. Given all of these results, it is possible that the deregulation of certain TFs, which is very likely to occur in the early stages of cancers [169,170,171,172], could result in RT changes and further contribute to cancer transformation. 

Genomic instability and, particularly, chromosomal instability could be another cause of RT changes (reviewed in [173]). Several early studies observed that chromosomal translocations were followed by changes in the replication timing of the translocated genes [174,175]. Thus, the juxtaposition of two regions with different replication timing may result in the earlier or later replication of at least one part of the translocated region (reviewed in [173]), presumably due to its insertion in a different chromatin and genetic environment. These early studies reported translocation events that cause RT changes at local domains, but more recent studies have revealed that certain chromosomal rearrangements can have a larger impact and affect the timing of a whole chromosome (reviewed in [173]). In this case, the translocation results in the disruption of *cis*-acting elements that regulate the RT program of individual chromosomes. Two examples have been described in particular: disruptions of the *ASAR6* and *XIST* genes result in extremely late replication of chromosomes 6 and X, respectively (reviewed in [173]). It has thus been proposed that all chromosomes could contain one specific locus that ensures an appropriate RT program for each chromosome (reviewed in [173]), whose disruption results in a chromosome-wide delay in RT. We could hypothesize that disruption of this crucial locus and the resulting large delay in chromosome replication could have a strong impact on cancer transformation (Figure 3).

Above, we describe different mechanisms that can lead to RT changes in the cancer context. However, to what extent do replication timing aberrations contribute to cancer progression? Do these RT changes trigger cancer transformation or are they consequences of alterations in counterpart processes? The answers to these questions remain unclear at the moment, as no direct relationship between RT changes and the cancerous phenotype has been confirmed yet. Nonetheless, given the interconnectedness of the replication timing with chromatin, transcription, and genomic stability, it is reasonable to suspect that RT changes induce abnormal variations in these three processes and contribute to or even amplify the development of cancer. 

## 9. Consequences of DNA Replication Timing Variations in Cancers

Replication timing changes may alter the chromatin structure [176,177]. An early study demonstrated that the early vs. late replication timing differentially dictated the chromatin state. By introducing exogenous gene constructs at different time points during the S-phase, Zhang et al. found that plasmid DNA introduced in early and late S-phase nuclei were packaged into the acetylated and deacetylated chromatin, respectively [176]. A subsequent study from the same lab further demonstrated that plasmid DNA injected during the late S-phase (thus containing deacetylated histones) became remodeled with acetylated histones when undergoing a round of early replication, and vice versa [177]. These results show that the timing of replication within the S-phase is involved in the regulation of the acetylation state of newly synthesized chromatin. Taken together, these results strongly suggest that RT shifts can affect the acetylation status of histones and therefore precede changes in epigenetic modifications.

Replication timing may also affect gene expression profiles. We previously described the potential impact of variations in transcriptional activity on RT. Conversely, replication changes may also affect the transcription competence of a region. Some but not all cancer-related genes associated with significant changes in RT were found to also display significant changes in their expression, either increasing or decreasing it [32]. It remains unclear how RT changes influence transcriptional activity, but one suggested mechanism is the modification of the chromatin landscape. For example, one recent study found that a small part of the genome showed differences in RT between normal and prostate cancer cells and was associated with transcriptional and chromatin variations [2]. Among these regions, they found that a number of genes replicated later in cancer cells and were associated with repressed expression and the repressive epigenetic status of the promoter. A gene set enrichment analysis (GSEA) showed that these genes were enriched in terms related to cancer (EMT, TNF-alpha signaling, KRAS signaling, cell movement, and cell proliferation) and tumor suppressor genes. However, these studies only present associations between RT changes and transcription variations. As mentioned previously, deciphering the causal relationship between RT and transcriptional activity remains complex.

Replication timing changes may eventually lead to replication stress. Local variations in RT may result in discoordination of the replication and transcription machineries, thus promoting collisions. Moreover, an excess of origin licensing at the same time may lead to a lack of factors necessary for the replication process (such as nucleotides or replication proteins) and impede global replication progression, resulting in the induction of replication stress [178,179]. Experiments testing the impact of RT aberrations on replication stress remain to be performed. This replication stress may in turn result in genomic and chromatin instability, which participates in cancer transformation.

Replication timing changes may also induce genomic/chromosomal instability. As described previously, replication timing seems to have direct consequences on mutagenesis and thus may be a contributor to genomic instability, a major hallmark of cancer (Figure 3). In addition, several studies have highlighted that RT may promote chromosomal instability [2,134] in two ways. First, some of these studies have reported that the RT status of a genomic region may play a role in chromosomal rearrangements. For example, several oncogenes involved in the formation of fusion genes due to chromosomal translocations have been found to be located in or near TTRs in the 11q and 21q chromosomal regions [180]. In particular, these cancer-related genes were present in GC-rich regions near regions of GC content transition and within regions of the early/late RT switch. Their results suggest that TTRs coincided with “unstable” regions that could be involved in the genomic instability of these nearby regions. The RT status of a locus may also influence the nature of chromosomal rearrangements. Indeed, analyses of prostate and breast cancers have shown significant enrichments of intra-chromosomal rearrangements (i.e., *cis* rearrangements) at late-replicating regions and inter-chromosomal rearrangements (i.e., *trans* rearrangements) at early-replicating regions [2]. For this purpose, the authors performed combinatorial epigenomic and RT analyses in prostate and breast cancer cells and found altered chromatin states at later- and earlier-replicating regions compared to in normal cells. Thus, they proposed an interesting model to explain why this specific chromosomal instability may be related to RT in cancers. The combination of the RT status, the spatial positioning and associated epigenetic alterations may influence the nature of chromosomal cancer rearrangements of early- and late-replicating loci. Second, different works have shown that RT changes correlate with chromosomal rearrangement sites [2,134,181]. By analyzing clinical prostate cancer datasets, Du et al. identified a number of chromosomal breakpoints that showed significant RT differences [2]. Similarly, through a whole genome-analysis of leukemic tumors, Ryba et al. found that a large shift from late to early replication was localized at a common site of translocation in leukemia (the *ETV6*/*RUNX1* translocation site) [134]. However, every leukemic cell type had this replication timing shift, but only a subgroup of them had the *ETV6*/*RUNX1* translocation, suggesting that the RT change preceded (and may have predisposed) this translocation.

## 10. Discussion and Conclusions

As described in the first paragraphs, RT is one of the main molecular elements of genome organization, and to date, no study has reported more than a rather low threshold of genome-wide RT changes in the studied genomes (Table 1). Similarly, our lab has performed numerous RT analyses in various conditions that challenge replication, and changes that exceed these limits have never been observed (unpublished data).

There are two possible explanations for the lack of such observations. Primarily, the RT program is a highly regulated process that makes it very robust. An alternative hypothesis is that cells cannot survive drastic RT changes, explaining the absence of such observations in studied cell populations. In conclusion, RT is extraordinarily and precisely reproducible, conserved from one species to another and, to some extent, adaptable to different circumstances. To our knowledge, no other molecular mechanisms display these same characteristics. Thus, RT shows very strong links to transcription, the chromatin state, genomic and nuclear organization, cell fate, evolution, etc. However, even if these links are very strong, there are still many remaining questions such as: (i) What are the molecular actors that control and regulate RT? and (ii) How can the “chicken-and-egg” paradox between these different processes be resolved? In the future, the molecular processes that facilitate the regulation of RT and its coordination with other nuclear processes must be unraveled and understood. This is one of the main challenges that must be carried out in the forthcoming years by different research teams. The combination of genomic, genetic, and biochemical approaches will help to achieve this task. However, new avenues of research can be explored such as the importance of nuclear pores and their components in the coordination of these processes, the identification of new post-translational modifications of different factors, and a more detailed characterization of nuclear territories that can be assimilated to “mini-factories” with “mini-stocks”.

The study of RT in the context of cancer is promising because, once again, strong links have been identified, and some results may lead to new therapeutic pathways. Indeed, a new treatment that results from an understanding of the link between RT and cancer and that targets an RT regulator could enable the specific inhibition of the proliferation of affected cells. However, much research is still needed to better understand the involvement of RT in the kinetics of oncogenesis. In cancer cells, RT variations have been shown to correlate with alterations in transcriptional activity, the chromatin state and genomic/chromosomal stability. The interconnectedness of the replication timing with these other nuclear processes [16,146,182] complicates the establishment of a timeline of events in tumorigenesis. One crucial question remains unresolved: Do aberrations in replication timing constitute an active process that alters all the other counterpart processes or are they a side effect of these alterations? Nonetheless, since the chromatin state, transcriptional activity, genomic stability, and the RT program are intimately connected, a change in one of these processes will surely impact the others, irrespective of the exact timeline of replication timing changes in malignant transformation. Thus, the participation of replication timing perturbations probably exacerbates the cancer phenotype through the deregulation of other crucial elements that shape the fate of the genome. Interestingly, it was very recently observed in human cancer cells that large advances in RT induced by replication stress in mother cells were also found in the daughter cells released from replication stress [29]. These specific genomic regions were associated with chromatin remodeling and gene upregulation in the daughter cells. For the first time, this study revealed that RT changes could be transmitted to the next cell generation with the persistence of alterations in chromatin structure and gene expression, which can then affect the fate of cancer cells.

To further clarify the relationship between RT and cancers, the development of cell models that perfectly characterize the onset of the expression of associated oncogenes would be valuable and permit analysis at the moment when RT is affected during its set-up. We could thus identify the first process disturbed—replication, transcription, epigenetic landscape or 3D organization of the genome—during these initial steps and thus understand the critical level of oncogenesis and genetic instability, which could be targeted by future therapeutic approaches.

Furthermore, single-cell approaches to studying RT [183,184] such as those being performed using patient tissues will refine the targets of therapeutic approaches and greatly improve personalized medicine, given the importance of RT in cancers.

## Figures and Tables

**Figure 1 ijms-22-04764-f001:**
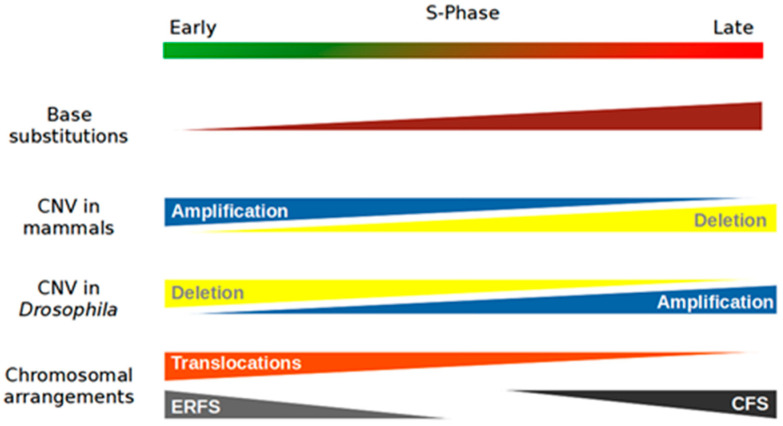
Schematic connections between mutational events and replication timing in eukaryotes. The height of the gradients is proportional to the observed probability of the corresponding mutational event. CNV: copy number variation; ERFS: early replicated fragile site; CFS: common fragile site.

**Figure 2 ijms-22-04764-f002:**
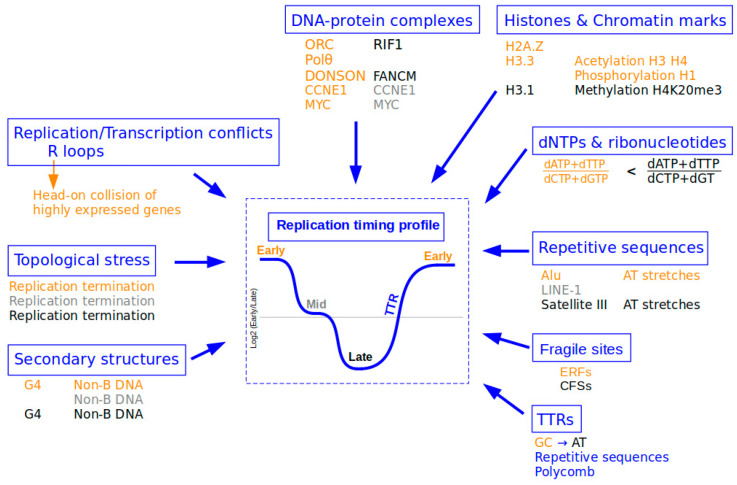
Interconnections between molecular features associated with replication timing and/or replication stress. Features specifically associated with TTRs and early-, mid- and late-replicating domains are depicted in blue, orange, gray, and black, respectively. TTR: timing transition region; G4: G-quadruplex; CNV: copy number variation; ERFS: early replicated fragile site; CFS: common fragile site.

**Figure 3 ijms-22-04764-f003:**
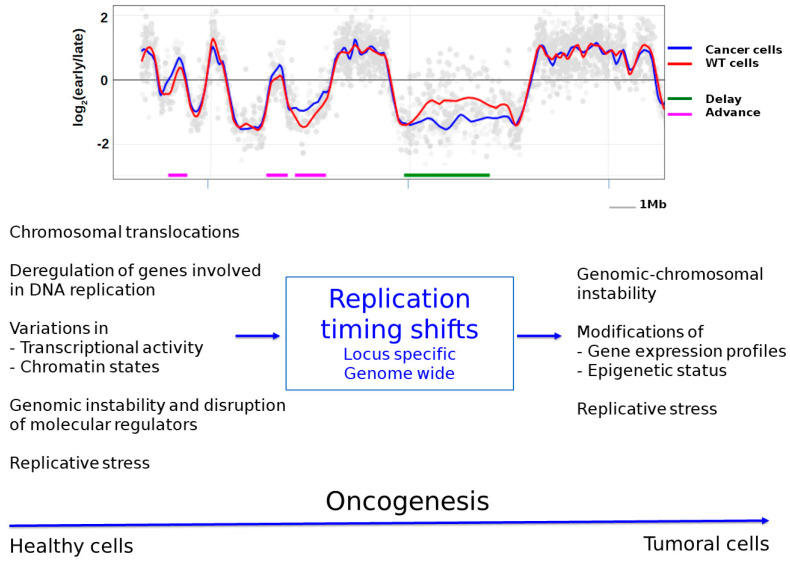
Links between replication timing, chromatin instability, and cancer. A typical RT profile is depicted above a diagram illustrating relations between RT shifts and genomic changes that potentially transform healthy cells into tumoral ones.

**Table 1 ijms-22-04764-t001:** Different studies reporting genome-wide RT changes.

	Percentage of RT Changes	Conditions	References
Drugs	1.5–6.5%	Low doses of APH in several cell lines (HCT116, RKO, U2OS, K562, MRC5-N and RPE-1)	[29]
	~4%	Low doses of APH in BJ-hTERT cells	[30]
Cancers	9–18%	17 pediatric leukemia tumors	[134]
	5.7%	Prostate cancers	[2]
Depletion of molecular RT regulators	25%	*PREP1* siRNA	[28]
	16.15%	*RIF1* KO	[10]
	15%	*SUV420H1* KO	[11]
	~5%	*POLQ* siRNA	[12]

APH: aphidicolin; *PREP1*: Pbx-Regulating Protein-1; *RIF1*: Replication Timing Regulatory Factor 1; *SUV420H1*: Su(Var)4-20 Homolog 1; *POLQ*: DNA Polymerase Theta.

## Data Availability

Not applicable.

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
