# Peer review of "Replication Stress, Genomic Instability, and Replication Timing: A Complex Relationship"

_ijms, 2021, doi:10.3390/ijms22094764_

Round 1

Reviewer 1 Report

Briu et al., Replication stress, genomic instability and replication timing: a complex relationship

In this review manuscript, Briu et al. described the complex relationship among replication timing, replication stress, and genomic instability in detail. Even though there is currently no consensus connecting the aforementioned fields, the authors summarized various findings surrounding these topics. In short, after a brief introduction (section #1), the authors explained the robustness of replication timing (#2), followed by the correlation between replication timing and genomic instability (#3). Then, they described the types of vulnerable regions of the genome upon replication stress such as CFS and ERFS (#4), and also discussed different sources of replication stress (#5), including their impact on replication timing (#6). After this, the authors focused on the replication timing changes in cancer cells (#7), discussed their causes (#8), and consequences (#9). In the end, they provided a future outlook for the relationship between replication timing and genomic instability (#10).

Overall, the article is well written, very thorough, and interesting to read. However, many sentences lack citations, especially in the first half of the manuscript. Also, some places were confusing. I recommend this review article for publication in IJMS after addressing the following points.

  1. One of the biggest problems was the lack of citation for so many sentences. It almost looks as if two different authors wrote the manuscript; one who doesn’t cite properly (first half) and one who does (second half). Below are places where I felt the citations were missing: Lines 33-36, 40-42, 60-61, 61-63, 70-71, 71-74, 74-75, 81-90, 92-93, 109-110, 103-106, 106-107, 109-110, 113-115, 117-124, 127-128, 137, 137-140, 146-147, 149-150, 164-166, 209-212, 216-217, 220-222, 244-245, 278-282, 315-317, 357-360, 673-676.

  1. Lines 52-58: I have never heard of this strange definition of chromosome instability and I strongly recommend deleting this because I think it is very confusing to redefine a term that is as prevalent as this. The DNA replication stress definition is fine but I felt that this is abrupt and a bit out of place.

  1. Lines 117-119: CTRs are similar to A/B compartment domains, not TADs. See refs #9, #15, #16. Note that ref #10 claimed that RT-switching domains correspond to TADs, not CTRs.

  1. Lines 137-140, 30-35%: When you give specific numbers, there is always a danger that these numbers could spread on their own without a firm ground. And these numbers can change easily depending on where you set your threshold. Either avoid them or give specific citations so that the readers know exactly what you are talking about.

  1. Lines 194-195: I could not understand the meaning of this sentence. Please clarify.

  1. Lines 212-213: In fission yeast, heterochromatic regions such as the pericentromeres and the silent mating-type locus are replicated in early S-phase (Kim et al., Genes Dev 2003, PMID: 12569122). Maybe change to “replicated generally in late S-phase”?

  1. Lines 259-263, In this context, I think it is necessary to introduce the Mitotic DNA synthesis (MiDAS) phenomenon and reference the relevant papers here. It is odd not to see MiDAS discussion at all.

  1. Lines 353-598, Chapter 5: This is an extremely lengthy chapter. The authors might want to consider subdividing it into several chapters?

  1. Lines 393-397 completely contradict the argument made on lines 355-396. Please explain.

  1. Lines 430-431, the authors could remove the paragraph break.

  1. Lines 433-434 is contrary to the argument made in line 431-442.

  1. Lines 478-493 is a duplicate of lines 457-472.

  1. Lines 494-499 is a duplicate of lines 473-478.

  1. Lines 617-620: Sarni et al. article (Nat Commun, 2020; PMID: 32680994) should be mentioned along with ref #102 as advanced replication in response to replication stress is discussed in this article.

  1. Lines 668-670: The authors say that RT perturbation causes genomic instability to induce tumorigenesis but nobody has shown this so this is misleading. Causality should be discussed very carefully.

  1. Line 917: Dileep and Gilbert article (Nat Commun, 2018; PMID: 29382831) should be mentioned along with ref #168.

Minor points

English grammatical mistakes here and there. For example:

  • Line 20, “deeply” is grammatically incorrect
  • Line 38, “whenever” –> “when”
  • Line 73, “as” –> “such as”
  • Line 92, “KO and KI” –> Were they spelled out?
  • Line 144, “It is therefore obvious that RT can be considered” is odd (obvious & can be in the same sentence)
  • Line 168, “normally” is unnecessary
  • Line 201, maybe change “besides” to “but”
  • Line 216, “IPSC cells” –> “iPS cells”
  • Line 277, maybe change “Thus” to “But” or “However”
  • Line 303, “(Barlow et al. 2013)” is unnecessary
  • Line 388, “DNA RT” –> “DNA RT regulation”
  • Line 613, “forward” –> “toward”
  • Line 1312, the title of ref #158 is incorrect

Author Response

We would like to thank you the reviewer #1 for their relevant comments that will definitely improve the quality of our review.

We used the "Track Changes" function in Microsoft Word to modify the manuscript according to the reviewers’ suggestions. We hope the changes will be clearly visible to the editors and reviewer. We took great care to address in this letter the issues the reviewer raised.

The line numbers were initially those of the pdf file of the submitted manuscript, they now correspond to the numbers of the pdf file of the revised version of the manuscript.

In addition to the revisions suggested by the reviewer, we did some other minor corrections listed below:

General English corrections:

Lines 97, 128, 149, 161, 176, 179, 182, 199-201, 205-206, 208, 225, 227, 233, 239, 241, 244, 262, 266, 268, 304-305, 338, 413, 522, 611, 652, 683, 711, 759, 761, 768, 791, 801, 838, 841, 846, 854, 857, 872-873, 890-891, 896, 909, 912, 916, 933, 962

General topological corrections:

Lines 74, 75, 92-93, 100, 124, 126, 174, 218-219, 231, 240, 266-267, 274, 275, 279, 301, 327, 338-339, 345-346, 366, 406, 576, 587, 614, 619, 682, 689, 690, 692

Sentence modifications and additional precisions:

Lines 76, 106-107, 110, 114, 295, 297, 298, 307, 310, 356-360, 364-365, 371-375, 397, 501, 576, 632-634, 666-670, 813-816

Other non-detailed revisions may result from the English editing service.

We also wanted to inform the editors that since we reorganized all the references (with Zotero), the formatting changes potentially done by the editorial office may have been lost.

The Reviewer #1 raised several major points:

  1. We thank the reviewer for bringing our attention on the point that some citations were lacking in the manuscript. We corrected this point by adding citations in brackets at the places where they were missing. We listed the location of the added citations at their new line numbers in the revised manuscript with their new corresponding numbers in brackets.

The citations are located at lines 36-39 [1], 43-45 [2], 67-70 [3], 76-79 [1,4], 79-81 [5], 81-83 [6], 89-98 [7], 100-104 [10-12], 117-119 [15], 120 [11], 123-124 [10, 17-20], 124-125 [12], 127-129 [22], 130-132 [16,23], 132-137 [16, 23-25], 139-140 [23, 26], 145-147 [23], 150-151 [27], 157-158 [20], 160-163 [10-12, 28-30], 169-171 [33], 173-174 [34], 236-238 [49], 243-245 [50,51], 247-249 [53], 317-320 [67-69], 400-402 [82,83], 632-633 [117], 739-742 [1].

  1. We agreed with the reviewer that our definition of chromosome instability is confusing so we deleted this sentence (lines 60-62 of the revised version, originally lines 52-54).

As also mentioned by the reviewer, the DNA replication stress definition was abrupt and a bit out of place. Thus, we rephrase this sentence (originally lines 55-58), tried to render it less abrupt and put it above in the introduction (lines 54-57 of the revised version). We changed the paragraph into:

We will then investigate the potential relationship between replication timing and DNA replication stress. DNA replication stress represents the alterations in the velocity of the ongoing replication forks, resulting in asymmetry in the fork progression, unreplicated regions and/or fork collapse. We will finally discuss the link and the impact of a deregulation of this temporal program on chromatin stability and consider the causes and consequences of the replication timing changes in the context of tumorigenesis.”

  1. The reviewer pointed out that we had a misleading interpretation of CTRs, A/B compartments and TADs (lines 117-119 of the initial submitted version of the manuscript). We tried to precisely rewrite this paragraph including references [9,10,15,16] now renumbered [23,26,24,25]. The modified version of this paragraph is the following (lines 132-147 of the revised text):

The CTRs consist of one or several Replication Domains (RDs) replicating at similar times. They are separated into two groups corresponding to A compartment, which is the active form and localizes with early replication domains, and B compartment, which represents the inactive form and is associated with late replication domains. A and B compartments likely represent euchromatin and heterochromatin compartments, respectively [16,23–25]. RDs are composed of one or several Topological Associated Domains (TADs). Some of them are closely located at compartment boundaries, corresponding to RT-switching [23,26]. Thus, TADs and RDs, that seem intimately linked, can be considered as the arbitrary units of the organization and the structure of a genome.”

  1. We thank the reviewer for this very relevant comment and agree there is a risk to give such numbers without clear explanations. Our idea was to point out that genome-wide RT changes have never been observed to exceed a defined limit around 30-35% according to many articles and our own observations. To avoid any confusion, we removed these numbers (originally located at lines 91-93, 96-97, 137-140, 861-863) and replaced them by more general terms such as “a rather low limit”, “a limited threshold”, “a limited percentage of RT changes” or “a rather low threshold of genome-wide RT changes” (lines 100-102, 108-110, 160-163, 930-932 in the revised manuscript). We inserted specific citations for illustrating our words (references [10-12] at lines 102-104).

We also decided to design a table resuming a set of studies that report accurate numbers of genome-wide RT changes (Table 1 at the beginning of the Discussion-Conclusion part, line 939 of the revised text). We think this table will clearly explain our idea and let the reader build its own interpretation.

  1. We thank the reviewer for bringing our attention on this point. We deleted this sentence (originally lines 194-195 and lines 220-221 in the revised manuscript) because we felt it was abrupt as it did not provide further information at this point.

  1. We thank the reviewer to point out this information. As suggested, we changed the sentence (originally lines 212-213 and lines 238-240 in the revised manuscript) to: “Heterochromatin must therefore potentially have more base substitutions than euchromatin, as heterochromatin is generally replicated in late S-phase.”

We also precised that the cited work deals with metazoans (originally lines 209-212 and lines 236-238 in the revised manuscript).

  1. As relevantly suggested by the reviewer, we described the Mitotic DNA synthesis (MiDAS) phenomenon (lines 286-295 in the revised text) and added related references [63-64].

The text we added is the following:

Thereby cells experiencing DNA replication stress may exhibit Mitotic DNA Synthesis (MiDAS) [63]. To understand the physiological function of MiDAS and its relationship to CFSs, the genomic sites of MiDAS were mapped in cells treated with aphidicolin (APH, an inhibitor of replicative DNA polymerases). Sites of MiDAS were evidenced as well-defined peaks that were largely conserved between cell lines and encompassed all known CFSs. The MiDAS peaks mapped within large, transcribed, origin-poor genomic regions. In cells that had been treated with APH, these regions remained unreplicated even in late S phase. MiDAS is then a rescue process that may help to complete replication in these regions [64].”

  1. The reviewer thought that the Chapter 5 was an extremely lengthy chapter that should be subdivided into several ones. We totally agree with this point of view. As advised by the reviewer, we subdivided this chapter in three parts.

5.1. Variations of the DNA components (lines 395-501 of the revised document)

5.2. Presence of proteins on DNA or chromatin (lines 503-595)

5.3. Existence of peculiar DNA structures (lines 597-662)

We also moved a paragraph concerning the FACT and histone H1 proteins from the 5.3. section to the 5.2. section (lines 565-575 of the revised text), as it appeared to us more relevant to the subdivided sections of Chapter 5.

  1. The reviewer noticed that the sentences located initially at lines 393-397 completely contradict the argument made on the all paragraph (originally lines 355-396).

To explain more clearly what we intended to say, we first added the following general sentence at the beginning of the paragraph (line 396 of the revised document).

In a physiological context, the cellular dNTP pool is finely regulated with specific concentrations for each nucleotide [80].”

We also changed “a balance supply of dNTPs” to “an appropriate supply of dNTPs during S-phase” and “since an altered or imbalanced nucleotide pool” to “since an altered nucleotide pool” at the lines 399 and 432-433 of the revised document, respectively.

Secondly, we rewrote these contradictory sentences (lines 436-441 of the revised text), which become:

The relative proportion of each dNTPs (dATP, dCTP, dGTP, dTCP) in normal and cancer human cells has already been established [80]. In normal cells, the four dNTPs are present in vivo at different concentrations, with the dGTP concentration being much lower than the others. Conversely, in cancer cells, the whole dNTP pool is highly increased although the dGTP level remains lower compared to other dNTPs.”

  1. As suggested by the reviewer, we removed the paragraph break (originally lines 430-431 and now line 477 of the revised document).

  1. The reviewer pointed out that the sentence (originally lines 433-434) is contrary to the argument made in the paragraph (originally lines 431-442).

We did rephrase this sentence to make it clearer (lines 479-482 from the revised manuscript). It becomes:

A deoxyadenosine is found upstream from most abundant genomic ribonucleic cytosines and guanines, with a strong propensity for ribonucleotides to be incorporated in short-nucleotide repeats initially containing deoxycytosines and deoxyguanines.”

  1. We thank the reviewer for seeing that some duplicates of lines remain in the text. We apologized for this inconvenience and we fixed it. Lines 478-493 is a duplicate of lines 457-472 (originally numbered lines).

  1. We thank the reviewer again for seeing that some duplicates of lines do exist in the text and we fixed it. Lines 494-499 is a duplicate of lines 473-478 (originally numbered lines). We have finally deleted the 527-548 lines of the revised version of the document.

  1. We agree with the reviewer that the work from Sarni et al. (Nat Commun, 2020; PMID: 32680994) should be mentioned along with the reference 102. We acknowledged that other contributions are important to mention and we apologized for not citing this work. We added this citation numbered [30] along with the reference 102, numbered [29] at the lines 682-686 (initially lines 617-620) in the revised version of the manuscript.

  1. The reviewer pointed out that it has not been shown that RT perturbation causes genomic instability to induce tumorigenesis. We did not want to discuss the causality of this statement and only wanted this sentence to make the link toward Chapter 8 of the text (at initially lines 668-670 and lines 733-736 of the revised document).

We softened the sentences that become:

This also suggests that perturbations of only a little part of this temporal program may be enough to cause sufficient genome instability to induce or participate in tumorigenesis. This will be discussed in further details later in the review.”

  1. We agree with the reviewer that the work from Dileep and Gilbert (Nat Commun, 2018; PMID: 29382831) should be mentioned along with the reference 168. This contribution is important to mention so we added this citation numbered [184] along with the reference 168 newly numbered [183] at the line 995 (originally line 917) of the revised manuscript.

The Reviewer 1 raised also minor points:

As mentioned by the reviewer, we corrected English grammatical mistakes in the text:

  • We replaced “deeply” by “in-depth” (originally line 20 and now line 22 of the revised document).

  • We changed “whenever” into “when” (originally line 38 and now line 41 of the revised document).

  • We modified “as” into “such as” (originally line 73 and now line 81 of the revised document).

  • We spelled out “KO and KI” (originally line 92 and now line 100 of the revised document).

  • We modified “It is therefore obvious that RT can be considered” into “it is therefore obvious that RT is now considered as…” (originally line 144 and now line 168 of the revised document).

  • We removed the unnecessary word “normally” (originally line 168 and now line 194 of the revised document).

  • We replaced “Besides” by “Nonetheless” (originally line 201 and now line 227 of the revised document).

  • We modified “IPSC cells” into “iPS cells” (originally line 216 and now line 243 of the revised document).

  • The reviewer asked us to change the word “Thus” but we finally deleted this sentence that appeared not so relevant in this paragraph (originally line 277 and now line 315 of the revised document).

  • We removed the unnecessary words “(Barlow et al. 2013)” (originally line 303 and now line 341 of the revised document).

  • We replaced “DNA RT” by “DNA RT regulation” (originally line 388 and now line 431 of the revised document).

  • We changed “moving forward” into “entering into” (originally line 613 and now line 679 of the revised document).

  • We replaced the title of the reference 158 (now numbered [173]) that was incorrect by the correct one (originally line 1312 and now line 1385 of the revised documen

Reviewer 2 Report

The topic of this review is very important and timely, and comprehensive reviews of this area (replication timing, replication stress, genome instability, replication fork/transcription bubble collisions, etc.) are lacking. I therefore very much hope to see this review published. The content areas are appropriate and highly interesting. Unfortunately, I found the writing quite confusing throughout, and the entire manuscript needs a careful revision to ensure that points are made clearly and convincingly. It is not simply a matter of correcting grammar - it is simply difficult to understand the meaning of many sentences. Just one example, "What is the first event occurring during the process of genome instability among replication timing degregulation and changes in chromatin that can lead to the appearance of cancer cells." I hope the authors can revise the manuscript or work with someone who can, because I believe it merits publication; it just needs to be more clearly written.

Author Response

We would like to thank you the reviewer #2 for their relevant comments that will definitely improve the quality of our review.

We used the "Track Changes" function in Microsoft Word to modify the manuscript according to the reviewers’ suggestions. We hope the changes will be clearly visible to the editors and reviewer. We took great care to address in this letter the issues the reviewer raised.

The line numbers were initially those of the pdf file of the submitted manuscript, they now correspond to the numbers of the pdf file of the revised version of the manuscript.

In addition to the revisions suggested by the reviewer, we did some other minor corrections listed below:

General English corrections:

Lines 97, 128, 149, 161, 176, 179, 182, 199-201, 205-206, 208, 225, 227, 233, 239, 241, 244, 262, 266, 268, 304-305, 338, 413, 522, 611, 652, 683, 711, 759, 761, 768, 791, 801, 838, 841, 846, 854, 857, 872-873, 890-891, 896, 909, 912, 916, 933, 962

General topological corrections:

Lines 74, 75, 92-93, 100, 124, 126, 174, 218-219, 231, 240, 266-267, 274, 275, 279, 301, 327, 338-339, 345-346, 366, 406, 576, 587, 614, 619, 682, 689, 690, 692

Sentence modifications and additional precisions:

Lines 76, 106-107, 110, 114, 295, 297, 298, 307, 310, 356-360, 364-365, 371-375, 397, 501, 576, 632-634, 666-670, 813-816

Other non-detailed revisions may result from the English editing service.

We also wanted to inform the editors that since we reorganized all the references (with Zotero), the formatting changes potentially done by the editorial office may have been lost.

The Reviewer 2 raised a major point:

The reviewer found that all the manuscript needed a careful revision to make it more clearly written and more convincing. We have then carefully revised the whole manuscript and tried to clarify unclear sentences.

For example, we transformed “What is the first event occurring during the process of genome instability among replication timing deregulation and changes in chromatin that can lead to the appearance of cancer cells.” into “It is now well known that replication timing deregulation and changes in chromatin can lead to the appearance of cancer cells. However, one main question still remains open. What is the first event occurring during the process of genome instability?”. These sentences were originally at lines 42-45 and are now at lines 45-48 of the revised document.

We then tried to shorten or subdivide other unclear sentences throughout the manuscript as well. We tried to revise all the English corrections we could and we also decided to submit our review to the “English editing” service provided by the IJMS journal to improve the clarity of the English language and style.The English editing service changes are in red in the new submitted review.

Reviewer 3 Report

This is very well written comprehensive review that was a pleasure to read. I have a few minor comments listed below:

Line 73: change to: "such as nucleotides".

Line 119: What is the distinguishing topological feature that demarcates compartments A and B?

Line 176: The authors say “a number of reviews elaborate on them” but give only one reference. Please change as “is reviewed elsewhere”.

Lines 273-279: The authors claim replication-transcription conflicts cause R-loops leading to fork stalling which is correct. However, they then contradict their own statement in the next sentence that R-loops do not seem to play a major role in CFS instability without any further discussion as to why. This is discussed later in the manuscript but may confuse the reader at this point. Please address this.

Line 397: Did you mean the “dGTP” level is highly increased?

Lines 401-404: Early replicating regions are GC rich while late replicating regions are AT rich. Is the dNTP ratio a reflection of this?

Lines 456-500: Repeated information. Please delete duplicate sentences.

Line 632: change as “because they are not properly resolved”.

Author Response

We would like to thank you the reviewer #3 for their relevant comments that will definitely improve the quality of our review.

We used the "Track Changes" function in Microsoft Word to modify the manuscript according to the reviewers’ suggestions. We hope the changes will be clearly visible to the editors and reviewer. We took great care to address in this letter the issues the reviewer raised.

The line numbers were initially those of the pdf file of the submitted manuscript, they now correspond to the numbers of the pdf file of the revised version of the manuscript.

In addition to the revisions suggested by the reviewer, we did some other minor corrections listed below:

General English corrections:

Lines 97, 128, 149, 161, 176, 179, 182, 199-201, 205-206, 208, 225, 227, 233, 239, 241, 244, 262, 266, 268, 304-305, 338, 413, 522, 611, 652, 683, 711, 759, 761, 768, 791, 801, 838, 841, 846, 854, 857, 872-873, 890-891, 896, 909, 912, 916, 933, 962

General topological corrections:

Lines 74, 75, 92-93, 100, 124, 126, 174, 218-219, 231, 240, 266-267, 274, 275, 279, 301, 327, 338-339, 345-346, 366, 406, 576, 587, 614, 619, 682, 689, 690, 692

Sentence modifications and additional precisions:

Lines 76, 106-107, 110, 114, 295, 297, 298, 307, 310, 356-360, 364-365, 371-375, 397, 501, 576, 632-634, 666-670, 813-816

Other non-detailed revisions may result from the English editing service.

We also wanted to inform the editors that since we reorganized all the references (with Zotero), the formatting changes potentially done by the editorial office may have been lost.

The Reviewer 3 raised some points:

  1. This point raised by the reviewer is similar to one minor point raised by the reviewer 1. We then modified “as” into “such as” (originally line 73 and now line 81 of the revised document).

  1. The reviewer 3 raised the same concerns as the reviewer 1 (on its major point 3) and asked us to clarify the notion of A/B compartments and TADs. We agreed that this paragraph lacked some information and we tried to precisely rewrite it. 

    1. The reviewer pointed out that we had a misleading interpretation of CTRs, A/B compartments and TADs (lines 117-119 of the initial submitted version of the manuscript). We tried to precisely rewrite this paragraph including references [9,10,15,16] now renumbered [23,26,24,25]. The modified version of this paragraph is the following (lines 132-147 of the revised text):

    The CTRs consist of one or several Replication Domains (RDs) replicating at similar times. They are separated into two groups corresponding to A compartment, which is the active form and localizes with early replication domains, and B compartment, which represents the inactive form and is associated with late replication domains. A and B compartments likely represent euchromatin and heterochromatin compartments, respectively [16,23–25]. RDs are composed of one or several Topological Associated Domains (TADs). Some of them are closely located at compartment boundaries, corresponding to RT-switching [23,26]. Thus, TADs and RDs, that seem intimately linked, can be considered as the arbitrary units of the organization and the structure of a genome.”

  1. As advised by the reviewer, we changed “a number of reviews elaborate on them” by “is reviewed elsewhere” as we gave only one reference in the text at that line (initially line 176 and now line 202 of the revised document).

  1. The reviewer evidenced two contradictory sentences. The first one claims that replication-transcription conflicts cause R-loops leading to fork stalling but the second one explains that R-loops do not seem to play a major role in CFS instability, which is contradictory. We thank the reviewer for bringing our attention on this point.

As this point is discussed later in the manuscript, we decided to delete this sentence (initially lines 277-279 and now 315-317 of the revised document).

  1. The reviewer wondered if the “dGTP” level is highly increased in cancer cells. This is not what we intended to say, so we clarified this point by changing the sentence “Conversely, in cancer cells, the dNTP pool is highly increased.” into “Conversely, in cancer cells, the whole dNTP pool is highly increased although the dGTP level remains lower compared to other dNTPs.” (initially line 397 and now lines 440-441 of the revised document).

  1. The reviewer asked if the dNTP ratio is a reflection of the early replication of CG-rich sequences and the late replication of AT-rich sequences. As far as we know, this notion remains unknown. We then added this sentence to highlight this point: “As far as we know, no direct relationship has been shown between GC-rich early replicating regions, AT-rich late replicating regions and the dNTP ratio. This remains to be explored in detail.” (initially lines 410-404 and now lines 446-448 of the revised document).

  1. The reviewer 3 noticed the same duplicated sentences than the reviewer 1 for its major points 12 and 13. We apologized for this mistake and we deleted the repeated sentences.

See our detailed answers to the major points 12 and 13 of the reviewer 1.

  1. We changed “because none properly resolved” into “because they are not properly resolved” (initially line 632 and now line 698 of the revised document).